

# Emissions of Carbon Tetrachloride (CCl$_4$) from Europe

Francesco Graziosi[1,2], Jgor Arduini[1,2,3], Paolo Bonasoni[3], Francesco Furlani[1,2], Umberto Giostra[1,2],
Alistair J. Manning[4], Archie McCulloch[5], Simon O'Doherty[5], Peter G. Simmonds[5], Stefan
Reimann[6], Martin K. Vollmer[6] and Michela Maione[1,2,3,*]

[1]Department of Pure and Applied Sciences, University of Urbino, Urbino, 61029, Italy
[2]CINFAI (National Inter-University Consortium for Physics of the Atmosphere and Hydrosphere), Rome,
00178, Italy
[3]Institute of Atmospheric Sciences and Climate, National Research Council, Bologna, 40129, Italy
[4]Hadley Centre, Met Office, Exeter, EX1 3PB, United Kingdom
[5]School of Chemistry, University of Bristol, Bristol, BS8 1TH, United Kingdom
[6]Laboratory for Air Pollution and Environmental Technology, Empa, Swiss Federal Laboratories for
Materials Science and Technology, Dubendorf, 8600, Switzerland

*Correspondence to*: Michela Maione (Michela.maione@uniurb.it)

**Abstract.** Carbon tetrachloride (CCl$_4$) is a long-lived radiatively-active compound able to destroy
stratospheric ozone. Due to its inclusion in the Montreal Protocol on Substances that Deplete the
Ozone Layer, the last two decades have seen a sharp decrease in its large scale emissive use with a
consequent decline of its atmospheric mole fractions. However, the Montreal Protocol restrictions
do not apply to the use of carbon tetrachloride as feedstock for the production of other chemicals,
implying the risk of fugitive emissions from the industry sector. The occurrence of such unintended
emissions is suggested by a significant discrepancy between global emissions as derived by
reported production and feedstock usage (bottom-up emissions), and those based on atmospheric
observations (top-down emissions). In order to better constrain the atmospheric budget of carbon
tetrachloride, several studies based on a combination of atmospheric observations and inverse
modelling have been conducted in recent years in various regions of the world. This study is
focused on the European scale and based on long-term high-frequency observations at three
European sites, combined with a Bayesian inversion methodology. We estimated that average
European emissions for 2006 -- 2014 were 2.3 (± 0.8) Gg yr$^{-1}$, with an average decreasing trend of
7.3 % per year. Our analysis identified France as the main source of emissions over the whole study
period, with an average contribution to total European emissions of 25%. The inversion was also
able to allow the localisation of emission "hot-spots" in the domain, with major source areas in
Southern France, Central England (UK) and Benelux (Belgium, The Netherlands, Luxembourg),
where most of industrial scale production of basic organic chemicals are located. According to our





results, European emissions correspond to 4.0 % of global emissions for 2006-2012. Together with other regional studies, our results allow a better constraint of the global budget of carbon tetrachloride and a better quantification of the gap between top-down and bottom-up estimates.

## 1.Introduction

Carbon tetrachloride ($CCl_4$) is a nearly exclusively anthropogenic compound whose first uses as solvent, fire extinguisher, fumigant and rodenticide date back to 1908 (Galbally, 1976; Happell et al., 2014). The rapid increase in its production occurring between the 1950s and the 1980s is linked mainly to its use as a solvent and also to the growth in the production of chlorofluorocarbons (CFCs) made from $CCl_4$ (Simmonds et al., 1998). This led to a significant increase in the

atmospheric mixing ratios of $CCl_4$, as shown by firn air analysis (Sturrock et al., 2002). The tropospheric lifetime, of 26 years  (SPARC 2013) to 35 years (Liang et al., 2014) is the result of the sum of three partial loss rates: loss in the stratosphere (Laube et al., 2013), degradation in the ocean (Yvon-Lewis and Butler, 2002) and degradation in the soil (Happell et al., 2014).

Main concerns about this long-lived chemical are linked to its capability to destroy the stratospheric

ozone layer and as a radiatively active gas, with an ozone depleting potential (ODP) of 0.72 (Harris and Wuebbles et al., 2014) and a global warming potential (GWP) of 1,730 (Myhre et al., 2013). The inclusion of $CCl_4$ in the Montreal Protocol on Substances that Deplete the Ozone Layer (MP) led to a sharp decrease in $CCl_4$ 's large scale emissive use and the consequent decline in its atmospheric mixing ratios was observed starting in the early 1990s (Fraser et al., 1994; Simmonds

et al., 1998), with peak mole fractions of around 103 part per trillion (ppt) and 101 ppt in 1991 in the Northern Hemisphere (NH) and Southern Hemisphere (SH), respectively (Walker et al., 2000). In 2012 $CCl_4$ measured global average mole fractions were 84.2 and 85.1 ppt, as measured by the AGAGE (Advanced Global Atmospheric Gases Experiment) and NOAA-GMD (National Oceanic and Atmospheric Administration-Global Monitoring Division) ground-based sampling networks,

respectively. The respective decrease rates during 2011-2012 were 1.2 and 1.6% yr$^{-1}$ (Carpenter and Reimann et al., 2014). The contribution of $CCl_4$ to total chlorine in the troposphere in 2012 was 10.3% (Carpenter and Reimann et al., 2014).

Currently, $CCl_4$ for uses that could lead to emissions is banned under the MP in all  countries. Production and use are allowed for feedstock for chemical manufacture, for example for

perchloroethylene,  hydrofluorocarbon (HFC) and pyrethroid pesticides production (UNEP, 2013). Chemical feedstocks should be converted into new chemicals, effectively destroying the feedstock, but fugitive emissions are possible. With no relevant natural emission sources (Butler et al., 1999; Sturrock et al., 2002) the possible sources for $CCl_4$ in the atmosphere are fugitive emissions from





the industry sector (Simmonds et al., 1998), generation during bleaching (Odabasi et al., 2014) or

emissions from a legacy of $CCl_4$ in old landfill (Fraser et al. 2014).

The persistence of such emissions is suggested by a discrepancy between global emissions as derived from reported production and feedstock usage (bottom-up emissions) and those based on atmospheric observations (top-down emissions). Assuming a total atmospheric lifetime of 26 years and the observed trend in the atmosphere, the top-down global $CCl_4$ emission estimates suggest for

2011-2012 global $CCl_4$ emissions are 57 (40–74) Gg yr$^{-1}$, a value that is at least one order of magnitude higher than estimates based on industrial use (Carpenter and Reimann et al., 2014).  In addition the persistence of an inter-hemispheric gradient of about 1.3 ppt (NH minus SH) since 2006, reinforces the hypothesis that $CCl_4$ is still emitted in the NH (Carpenter and Reimann et al., 2014).  Similar results have been obtained by Liang et al. (2014), who deduced that the mean global

emissions during 2000-2012 were 39 Gg yr$^{-1}$ (34-45 Gg yr$^{-1}$) with a calculated total atmospheric lifetime for $CCl_4$ of 35 (32-37) years.

In order to better constrain the $CCl_4$ budget, several top-down studies have been conducted in recent years focused on the global and regional scale, the top-down approach having been recognised as an important independent verification tool for bottom-up reporting (Nisbet and Weiss, 2010; Weiss

and Prinn, 2011; Lunt et al., 2015).

Xiao et al. (2010) used a three-dimensional inversion model and global $CCl_4$ observations (AGAGE and NOAA-GMD) to derive emissions from eight world regions over the 1996-2004 period, identifying South-East Asia as being responsible of more than half of the global industrial emissions, which they estimated as  74.1 ± 4.3 Gg yr$^{-1}$ (9-year average).

The role of China as a significant source region of $CCl_4$ has been highlighted by Vollmer et al. (2009) who, based on 18-month continuous high-frequency observations (October 2006 – March 2008) conducted at a site in the North China Plain and a Bayesian inversion modelling approach, calculated Chinese emissions to be 15 Gg yr$^{-1}$ (10-22 Gg yr$^{-1}$) out of their global estimates of 53 ± 30 Gg yr$^{-1}$.

According to Fraser et al. (2014) top-down Australian emissions during 1996-2011 have declined from 0.25-0.35 Gg yr$^{-1}$ to 0.12-0.18 Gg yr$^{-1}$. In this study potential sources other than those arising from production, transport and use were identified and on the basis of an analysis of pollution episodes, were likely to be associated with contaminated soils, toxic waste treatment facilities and chlor-alkali plants.

In 2012, Miller et al., used a $^{14}$C-based top-down method, to derive for the United States an average emission of 0.4 Gg yr$^{-1}$ during 2004-2009, corresponding to 4% of the global emissions given in Montzka and Reimann et al. (2011). The uncertainty inherent in estimating top down emissions is



highlighted by the work of Hu et al. (2016) who estimated the average emissions from United States during 2008-2012 to be 4.0 (2.0-6.5) Gg yr$^{-1}$. This number is two orders of magnitude greater that
emissions reported to the US EPA Toxic Releases Inventory over the same period and one order of magnitude larger that the previous estimates given by Miller et al. (2012). Hu et al.'s estimates were derived using observations from a large observation network including multiple sites across the United States and both a Bayesian and geostatistical inverse analyses.

For Europe, the most recent estimates are given in the above-cited paper by Xiao et al. (2010), who
reported that Europe has been responsible, over 1996-2004, for 4% of global emissions. However this study, based on observations conducted at global baseline sites, does not take into account regional variations that might occur across the different European countries and that could help in identifying specific emission sources, including those unrelated to reported production.

With this aim, we conducted a study based on long-term, high-frequency CCl$_4$ observations carried
out at three European sites combined with FLEXPART and the Bayesian inversion approach developed by Seibert (2000; 2001), improved by Eckhardt et al. (2008) and Stohl et al. (2009; 2010) and recently applied to derive emissions of halogenated species at the European scale (Maione et al., 2014; Graziosi et al., 2015).

Even though major source regions are likely to be located in East Asia, our results, in combination
with those obtained from other regional studies, are useful in order to better assess the global budget of CCl$_4$ and better evaluate to what extent future emissions will affect the evolution of the equivalent effective stratospheric chlorine (EESC).

## 2.Method

### 2.1 Measurements


In Europe CCl$_4$ long-term, high-frequency observations of CCl$_4$ are available from three sites, all labelled as WMO-GAW (World Meteorological Organisation-Global Atmosphere Watch) global stations and AGAGE and affiliated station: Mt. Cimone, CMN (Italy); Jungfraujoch, JFJ (Switzerland) and Mace Head, MHD (Ireland). CMN and JFJ are mountain stations occasionally
affected by air masses from the polluted boundary layer; MHD baseline station is mostly affected by clean oceanic air masses. All CCl$_4$ data used in this paper are available from the AGAGE network: different instrumentations and protocols are used to measure in situ CCl$_4$ at each station: CMN uses a gas chromatograph with mass spectrometric detection (GC-MS), with sample enrichment on adsorbent trap by a commercial thermal desorber (Maione et al., 2013); JFJ uses a
gas chromatograph with mass spectrometer detection, with sample enrichment on a custom built thermal desorber-Medusa-GC-MS, (Miller et al., 2008); MHD uses a gas chromatograph with an





electron capture detection (GC-ECD), without sample enrichment (Prinn et al., 2000). All the measurements are reported using the Scripps Institution of Oceanography (SIO), SIO-05 gravimetric primary calibration scale: ambient air measurements are routinely calibrated against whole air working standard that have been filled locally, using a bracketing technique, to override short term instrumental drifts. Working standards are then referenced on a weekly basis to tertiary tank (provided and calibrated by SIO) on site for the GC-MS measurements, i.e. CMN and JFJ. For the Mace Head GC-ECD instrument the tertiary tanks are used as the working standard are prepared and regularly calibrated at SIO at least twice, at the beginning and end of the life of the tank (Prinn et al., 2000; Miller et al., 2008). For this reason the contribution of the scale transfer (calibration) uncertainty to the total measurement uncertainty is minimized among stations, constraining the error estimate to the instrumental precision, calculated as the standard deviation ($1\sigma$) of the repeated working standard measurements for the covered period, that is typical for each site/setup and almost constant over the years of observation: CMN $\pm$ 0.39 ppt; JFJ $\pm$ 0.86 ppt and MHD $\pm$ 0.24 ppt. In addition, the analytical systems at the three stations are operated via the Linux-based chromatography software GCWerks (gcwerks.com) developed within the AGAGE programme.

**2.2 Inverse modelling**

Observations have been combined with 20-days backward trajectories of the FLEXPART Lagrangian Particle Dispersion Model (Stohl et al., 2005). FLEXPART runs are based on the European Centre for Medium-range Weather Forecast wind fields, using 3-hourly ERA-Interim reanalyses (analysis fields are at 00:00, 06:00, 12:00 and 18:00 UTC, and 3-h forecasts are at 03:00, 09:00, 15:00 and 21:00 UTC) with 1°x1° horizontal resolution and 91 vertical levels. The emission sensitivity map of source-receptor relationships (SRR) generated using the three European stations is reported in Fig.1. The obtained SRR combined with an *a priori* emission field allowed us to estimate the *a posteriori* emission flux for the European Geographic Domain (EGD), using the Bayesian inversion technique.

With the aim of obtaining the best performance of the model in terms of correlation coefficient between the observations and the modelled time series, we tested various *a priori* emission field settings. The best model performance was obtained using an *a priori* emission field built combining the emission fluxes estimated by Xiao et al. (2010) and those available at the European Pollutant Release and Transfer Register (E-PRTR). The E-PRTR is a Europe-wide register that provides environmental data from 30,000 industrial facilities covering 65 economic activities within 9 industrial sectors in EU member states, as well as in Iceland, Liechtenstein, Norway, Serbia and Switzerland and reports $CCl_4$ atmospheric emissions higher than 100 kg yr$^{-1}$. E-PRTR data are available from 2007 to 2013. More details on the E-PRTR are given in the Supplementary Material.



We built the *a priori* emission field as follows: the annual intensity emissions from 2006 to 2014 is obtained by applying to the European emission flux given in Xiao et al (2010) a decreasing rate of 2.2% per year (Carpenter and Reimann et al., 2014). The spatial emission distribution is achieved: i)

for the years 2006 and 2014 the flux was distributed onto the domain following the population density given by the Centre for International Earth Science Information Network (CIESIN 2010); ii) for the years from 2007 to 2013, the corresponding fraction of E-PRTR, on average 4.1% of total intensity *a priori* emission, was allocate following the geo referenced information of the same data base, while the remaining part of emission was disaggregated according to the population data

density as previously described.

The inversion grid consists of more than 5.000 grid boxes with different horizontal resolution ranging from 0.5° by 0.5° to 2.0° by 2.0° latitude-longitude in order to assure similar weight on the inversion result. We estimated nine years of European emissions, from January 2006 to December 2014. From January 2006 to December 2014 the inversion was run using the only two stations

(CMN and MHD) in which observations were available. During 2010-2014, also data from JFJ were used. A detailed description of the inversion technique and of the related uncertainty is given in the Supplementary Material.

### 3. Results and discussion.

### 3.1 Time Series Statistical Analysis

$CCl_4$ time series at three European stations are reported in Fig. 2. Using a statistical approach described in Giostra et al. (2011) we discriminate background mole fractions (black dots) from elevations above the baseline (red dots) due to pollution episodes. The monthly mean background mole fractions have been used to derive $CCl_4$ atmospheric trends, applying the empirical model

described in Simmonds et al. (2004). Atmospheric trends in the background mole fractions over the common period (July 2010- Dec 2014) are -1.5±0.2, -1.2±0.1, -1.3±0.1% yr$^{-1}$ ($R^2$ = 0.93, 0.99, 0.98), at CMN, MHD, and JFJ, respectively. Such values are consistent with global trends given in Carpenter and Reimann et al. (2014).

### 3.2 Inversion Results

$CCl_4$ emission intensity from the EGD and emission distribution within the same domain has been estimated using the European observations and the described Bayesian inversion technique. As shown in Figure 3, main deviations between our estimates (flux $_{post}$) and the *a priori* values (flux $_{prior}$) are found in 2006, and in 2013-2014. The relative percentage bias, given by (flux $_{post}$ − flux

$_{prior}$)/flux $_{prior}$ *100), ranges from + 25% to -32 %, as shown in the bottom panel of Fig 3. The



emission flux uncertainty decreases from 180% of the *a priori* to 35% of the *a posteriori* emission field (average over the study period), supporting the reliability of the results. More details on the method performance are given in the Supplementary Material.

### 3.2.1 European emissions and emission trends


The inversion results indicate average EGD emissions during study period of 2.3 (± 0.8) Gg yr$^{-1}$. $CCl_4$ total emissions from the EGD have decreased from ~ 3.1 (± 1.2) Gg yr$^{-1}$ in 2006 to ~ 1.5 (± 0.5) Gg yr$^{-1}$ in 2014, corresponding to an average EGD decreasing trend of 7.3 % per year. EGD and macro-areas emission estimates for the single years are given in Table I. Such figures cannot be

reconciled with potential emissions estimated from production data reported to UNEP that for Europe are negative along the study period with exception of 2012 (UNEP Production data base). Such discrepancy holds also if a 2% of fugitive emissions and a 75% of destruction efficiency are considered. Also when comparing our estimates with emissions from the industrial activities declared to the E-PRTR, we found the latter to be strongly (on average 35 times) under-estimated,

reinforcing the incompleteness of available information.

### 3.2.2 Emission distribution within the domain

The obtained EGD *a posteriori* emission fluxes differ from the *a priori* both in intensity (as described above) and in spatial distribution.

In order to quantitatively assess the contribution to the $CCl_4$ overall European emissions from the various countries, we have divided our domain into ten macro areas, whose extension is related to the SRR of the area (see Figure 1). Emissions from the single macro areas and the associated uncertainty (see Supplementary Material) are reported in Table 1 and in Figure 4a. Figure 4b shows the percentage contribution from the single macro areas.

Our estimates identify FR as the main emitter in the EGD all over the study period, with an average contribution of 25%. Five macro areas (ES-PT > NEE > DE-AT > UK-IE > IT) contribute between 14.5 and 9.1%, while the remaining regional contributions average 5% each. Emissions from France reached the maximum in 2010. Emissions from IT, CH and from NEE show a faster decreasing trend with respect to the average EGD rate and the remaining macro areas decreased according to

the overall average EGD emissions. As a result, starting from 2009, the percent contribution of France is about the 30 % of total EGD emissions.

### 3.2.3 Emission hot spots

Beside the overall picture given by the analysis of the aggregated macro area emission estimates,

the analysis of the spatial distribution of the emission fluxes provides additional insights. Maps in



Figure 5 show the *a priori* (Fig. 5a) and *a posteriori* (Fig. 5b) average distribution of emission fluxes over the study period. The geo referenced emission sources as reported by the E-PRTR inventory, subdivided in release to water (crosses) and release to the atmosphere (open circles) with the dimension of the circles referring to the amount released, are shown as well. The *a posteriori*

map shows a different distribution of the emission fluxes with respect to the *a priori*, and does not follow the population distribution. Figure 5b shows how, in general, the localisation of the main declared emission sources is well captured by the inversion, as in the case of Southern France, Central England (UK) and BE-NE-LUX. According to the E-PRTR inventory, these hot-spots account for more than 90% of the industrial scale production of basic organic chemicals. The hot

spots are observed even when the inversion is run using the *a-priori* emission field that does not include the E-PRTR information on industrial emissions, indicating that the emission hot spots are not forced by the *a priori* flux.

In order to facilitate the comprehension of the map in Fig. 5b, we compared the E-PRTR emission fluxes with estimates from the grid cells included in the corresponding hot spot areas identified

through the inversion. We found that emission fluxes for the hot spots in Southern France and Central England were one order of magnitude larger than the reported ones and for BE-NE-LUX emissions five times larger than those declared in the E-PRTR inventory. These results suggest either an under-reporting of current emissions and/or the occurrence of additional sources not reported by the E-PRTR inventory and/or emissions from historical production (such as landfill) or

chlor-alkali industry (Fraser et al., 2014).

### 3.2.4 Comparison with NAME

For comparison, we ran an alternative top down approach based on observations at MHD combined with the UK MetOffice Numerical Atmospheric dispersion Modelling Environment (NAME) to

simulate the dispersion and an iterative best fit technique (the simulated annealing) to derive regional emission estimates (Manning et al., 2011). This alternative top-down approach differs from our procedure in the dispersion model, in the inversion technique, in the absence of an *a priori* emission field and in the use of a single receptor. The use of a single station narrows the study area to a sub-EGD that includes eight countries in North West Europe (NWEU), i.e. BE-NE-LUX, DE,

DK, FR, IE and UK. Figure 6 reports a comparison of the results obtained using the two different approaches for UK only and for the NWEU domain. Overall, a fair agreement is observed, with the differences between the two estimates always within the emission uncertainty. Such encouraging results endorse the reliability of the estimated emissions.





### 3.3 The global perspective


To put European emissions in a global perspective, we compared our results with global estimates. Global top-down emissions as derived from atmospheric measurements are available only until 2012 (Carpenter and Reimann et al., 2014). For the sake of consistency, this comparison was made considering the same time period, when we estimated EGD average emissions of 2.5 Gg yr$^{-1}$,

corresponding to 4% of the global average. The plot in Fig.7 shows a comparison between the EGD and the global emission trends. Over 2006-2012, the EGD estimates show an average trend -3.9% yr$^{-1}$ compared with a global trend, for the same period, of -2.2% yr$^{-1}$. This suggests that European CCl$_4$ emission sources are weakening faster than the global ones.

Based on our inversion EGD per-capita emissions during 2006-2014 were 4.4 (3.0 – 5.9) g yr$^{-1}$

against a global per-capita flux of 9.0 (6.4 – 11.6) g yr$^{-1}$ in 2006-2012. Based on other regional emission studies we derived per-capita emissions of 7.2 (6.6 – 7.9) g yr$^{-1}$ (2004-2111) for Australia (Fraser et al., 2014), 11.7 (7.8 – 17.2) g yr$^{-1}$ in 2007 for China (Vollmer et al., 2009) and 12.9 (6.5 – 21.0) g yr$^{-1}$ during 2008-2012 for United States (Hu et al., 2016).

### 4. Conclusions


Average carbon tetrachloride emissions from the European Geographic Domain during 2006-2014 derived from atmospheric observations combined with a Bayesian inversion method have been estimated at 2.3 (± 0.8) Gg yr$^{-1}$, with a decreasing rate of 7.3% per year.

When comparing emissions derived with the top-down approach with those evaluated through bottom-up methods, large discrepancies are observed. Such discrepancy is expected with regard to the information contained in the UNEP database, which reports production (without allowing for stock change but quoting destruction as a negative production) and consumption for emissive uses. On the other hand, emissions reported in the E-PRTR inventory should include measurements or

estimates from industrial processes (including waste treatment) that can potentially emit CCl$_4$ but they represent only 3% of our emission estimates. In spite of the discrepancy in the quantification of emissions, the inversion is able to localise the main source areas reported in the E-PRTR, as also observed in the United States by Hu et al. (2016). Major source areas identified in the EGD are Southern France, Central England (UK) and BE-NE-LUX, where most of industrial scale

production of basic organic chemicals are located.

Diffusive emissions other than current industrial processes could be hypothesised. Domestic emissions due to the use of bleach-based cleaning agents have been estimated to be 20% of our



emissions for the EGD (Odabasi et al., 2014). Fraser et al. (2014) hypothesised that the emissions from Australia were from contaminated soil, toxic waste treatment and were coincident

geographically with chlor-alkali production (that is, the emissions were largely from historical industrial processes). Natural emissions have been found to be negligible (less than 5%) on a global scale (Butler et al, 1999; Sturrock et al., 2002). This suggests an underestimation of $CCl_4$ emissions from industry and/or the incompleteness of the register of declared sources and/or other sources, such as use of cleaning agents and contaminated landfills.

According to our estimates, the European contribution to global emissions of $CCl_4$ was 4% during 2006-2012 with a decreasing trend of 3.9% $yr^{-1}$ compared with a global trend of 2.2% $yr^{-1}$ for the same period. Note that when considering the whole study period (2006-2014), the estimated emission flux decreasing trend was, as reported above, 7.3% $yr^{-1}$. Europe exhibits the lowest per-capita emission, 4.4 (3.0 – 5.9) g $yr^{-1}$, among those areas where regional studies are available.

To summarize, this study allowed us to estimate $CCl_4$ emission fluxes for the European regional scale independently from inventories based on bottom-up procedures. Thanks to the good sensitivity in most of the EGD, the emission field can be reconstructed with a resolution level able to show, for each country, the main inconsistencies between the national emission declarations and the estimates based on atmospheric observations. Moreover, together with other regional studies

(Fraser et al., 2014; Hu et al., 2016; Vollmer et al., 2009) our results will allow a better constraint of the global budget of $CCl_4$ and a better quantification of the gap between top-down and bottom-up estimates.

*Acknowledgements.* We acknowledge the AGAGE science team as well as the station personnel for

their support in conducting in situ measurements. Measurements at Jungfraujoch are supported by the Swiss Federal Office for the Environment (FOEN) through the project HALCLIM and by the International Foundation High Altitude Research Stations Jungfraujoch and Gornergrat (HFSJG). Measurements at Mace Head are supported by the Department of the Energy and Climate Change (DECC, UK) (contract GA0201 to the University of Bristol). The InGOS EU FP7 Infrastructure

project (grant agreement n° 284274) also supported the observation and calibration activities. The University Consortium CINFAI (Consorzio Interuniversitario Nazionale per la Fisica delle Atmosfere e delle Idrosfere) supported F. Graziosi grant (RITMARE Flagship Project). The "O. Vittori" station is supported by the National Research Council of Italy.

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





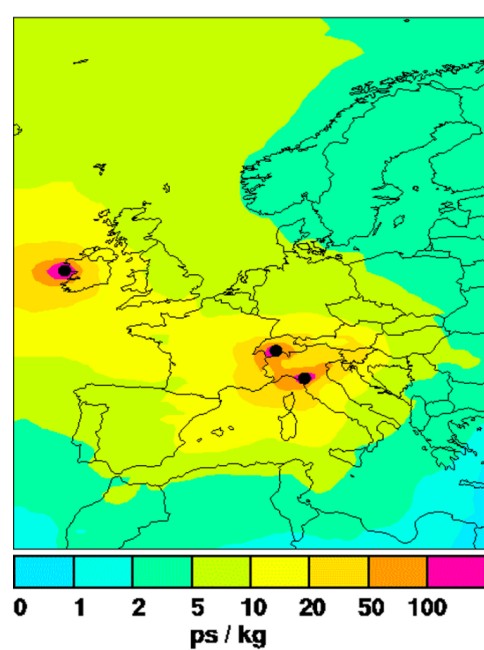

**Figure 1. Footprint emission sensitivity in picoseconds per kilogram (ps kg$^{-1}$) obtained from FLEXPART 20 days backward calculations averaged over all model calculations over two years (Jan 2008- Dec 2009). Measurement sites are marked with black dots.**



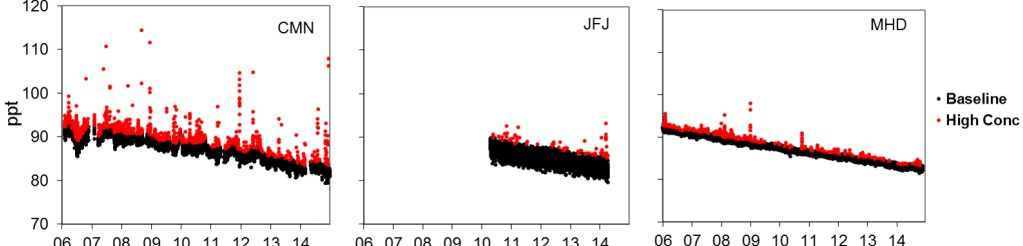

**Figure 2. CCl₄ time series at three European sites. Black dots: baseline, red dots: enhancements above the baseline.**





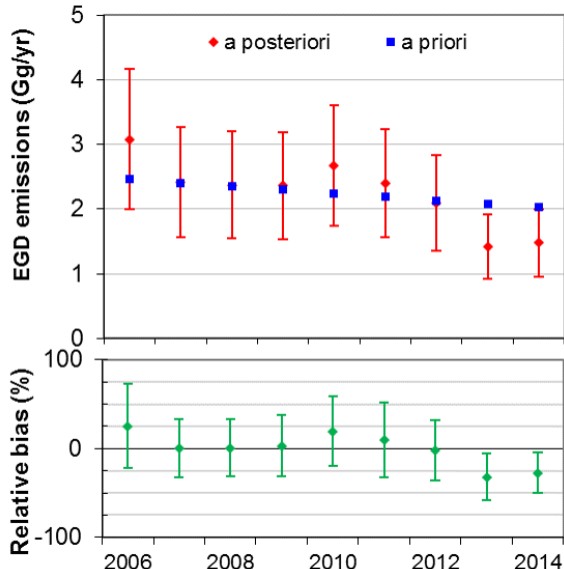

**Figure 3. Upper panel: comparison between the *a priori* (blue line) and *a posteriori* CCl$_4$ emission fluxes from the European Geographic Domain during 2006-2014. Bottom panel, percentage relative bias between the *a priori* and *a posteriori* time series.**




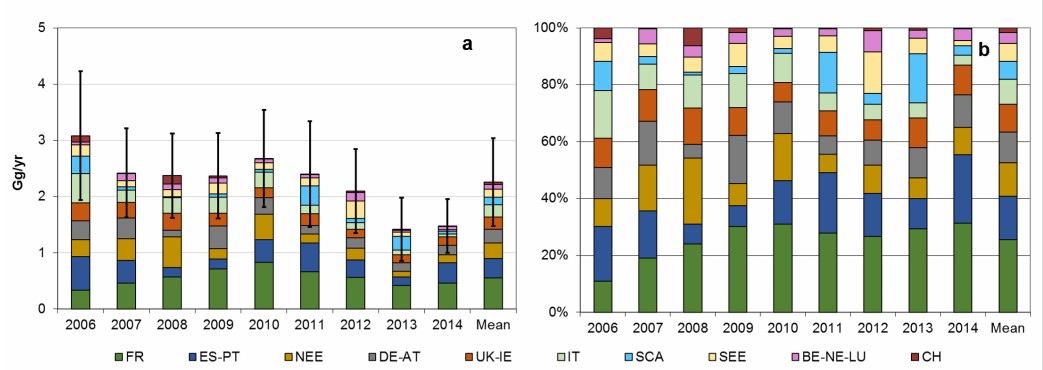

Figure 4. a) Carbon tetrachloride estimated emission over the study period given in Gg yr[-1] from ten macro areas in the EGD. b) Yearly percent contribution of the single areas to total EGD emissions







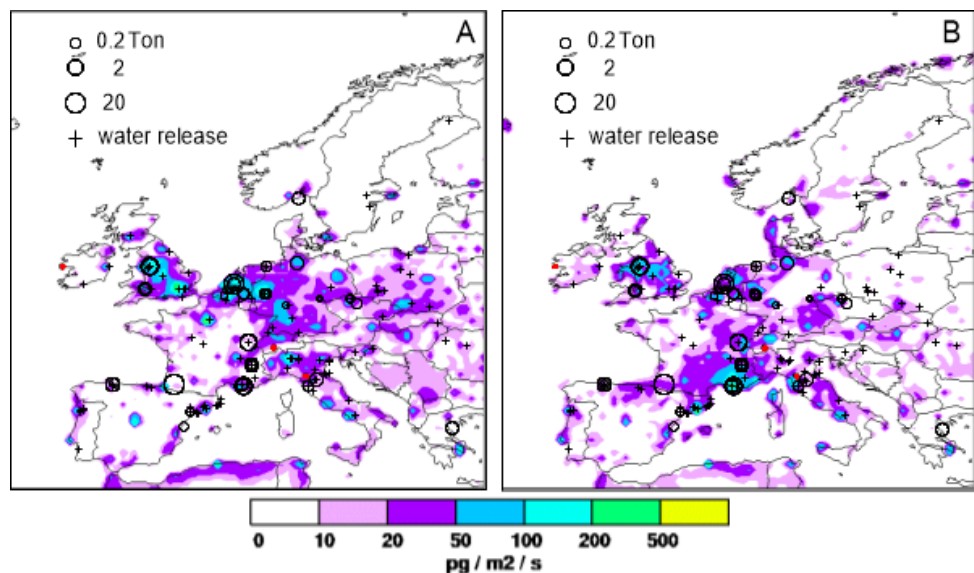

**Figure 5: a) Average *a priori* distribution of CCl₄ emissions from the European Geographic Domain over the study period; b) *a posteriori* distribution for the same domain. Measurement stations are marked with red dots. Open circles represent emissions to atmosphere as reported by the E-PRTR inventory and crosses correspond to release to water, as for the same inventory.**






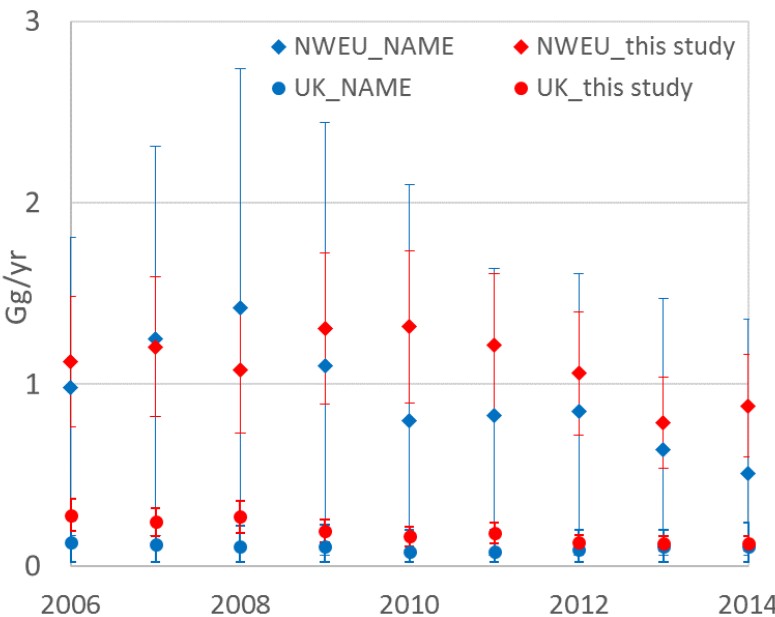

**Figure 6. Comparison between emissions from UK (circles) and the NWEU domain (diamonds) estimated through the NAME (blue) and the Bayesian (red) approach.**



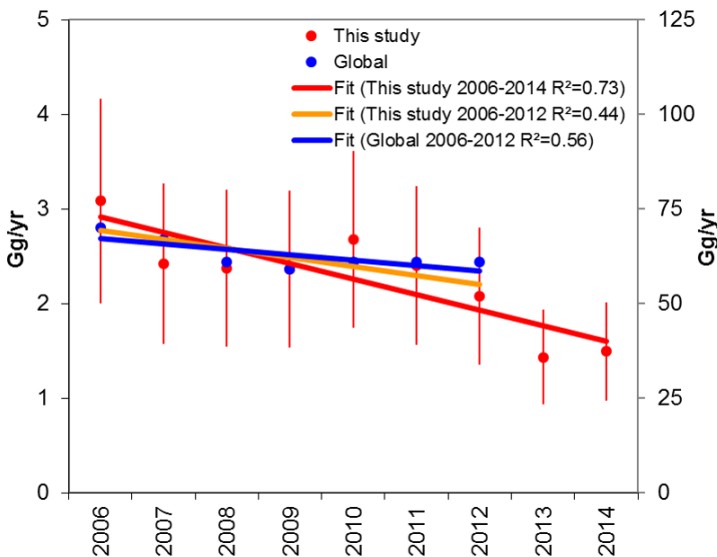

**Figure 7. Comparison between top-down emission trends in the EGD (this study), red lines left axis and the global ones (Carpenter and Reimann et al., 2014), blue lines right axis.**





**Table 1: Carbon tetrachloride emission estimates (Gg yr$^{-1}$) and associated uncertainty, percent yearly emission trends and 9-yr average percent contribution from the EGD and from ten macro areas in the EGD over the study period. Macro areas, listed according to their emission intensity are: FR (France); ES-PT (Spain, Portugal); NEE (Poland, Czech Republic, Slovakia, Lithuania, Latvia, Estonia, Hungary, Romania, Bulgaria); UK-IE (United Kingdom,**

**Republic of Ireland); DE-AT (Germany, Austria); IT (Italy); SCA (Norway, Sweden, Finland, Denmark); SEE (Slovenia, Croatia, Serbia, Bosnia-Herzegovina, Montenegro, Albania, Greece); BE-NE-LUX (Belgium, The Netherlands, Luxembourg), CH (Switzerland).**

| Areas | CCl$_4$ yearly emissions (Mg yr$^{-1}$) | | | | | | | | | trend | % |
|---|---|---|---|---|---|---|---|---|---|---|---|
| | 2006 | 2007 | 2008 | 2009 | 2010 | 2011 | 2012 | 2013 | 2014 | %yr$^{-1}$ | mean |
| EGD | 3082 ±1152 | 2417 ±790 | 2370 ±747 | 2366 ±761 | 2676 ±867 | 2397 ±933 | 2096 ±731 | 1416 ±557 | 1476 ±471 | -7.3 | contrib ution |
| FR | 336±91 | 459±124 | 567±153 | 713±193 | 831±224 | 665±180 | 560±151 | 416±112 | 462±125 | 0.4 | 25 |
| ES-PT | 591±215 | 400±146 | 167±61 | 172±63 | 405±148 | 511±186 | 315±115 | 150±55 | 355±129 | -6.3 | 14.5 |
| NEE | 304±115 | 390±148 | 551±209 | 186±70 | 447±169 | 155±59 | 211±80 | 103±39 | 143±54 | -13.1 | 12 |
| DE-AT | 336±94 | 372±104 | 115±32 | 401±112 | 294±82 | 157±44 | 184±51 | 151±42 | 168±47 | -8.9 | 11.1 |
| UK-IE | 316±79 | 271±68 | 303±76 | 230±57 | 182±45 | 207±52 | 151±38 | 148±37 | 154±38 | -8.6 | 10.3 |
| IT | 520±149 | 218±62 | 273±78 | 284±81 | 276±79 | 152±44 | 114±33 | 77±22 | 51±15 | -20.5 | 9.1 |
| SCA | 314±258 | 62±51 | 24±20 | 60±49 | 45±37 | 343±282 | 78±64 | 244±200 | 49±40 | -1.3 | 6.3 |
| SEE | 205±120 | 110±65 | 125±73 | 192±113 | 117±69 | 141±83 | 308±181 | 77±45 | 26±15 | -7.1 | 6 |
| BE-NE-LU | 45±8 | 128±21 | 95±16 | 90±15 | 70±12 | 57±10 | 155±26 | 39±7 | 62±10 | -0.6 | 4 |
| CH | 115±23 | 7±1 | 150±30 | 38±8 | 9±2 | 8±2 | 22±4 | 12±2 | 6±1 | -28.6 | 1.6 |
