# Peer review of "Emissions of Carbon Tetrachloride (CCl4) from Europe"

_Atmospheric Chemistry and Physics, 2016_

## Referee Comment (RC1) · P. J. Fraser (Referee) · 30 May 2016

Text suggestions

Line 40: near exclusively

Line 45: (Butler et al., 1999; Sturrock et al., 2002)

Line 53: sharp decrease in the large-scale emissive use of CCl4

Line 63: emissive uses of CCl4 are banned...in signatory countries

Line 67: no significant natural sources

Line 69: the industry sector (Simmonds et al., 1998; Fraser et al., 2014)

Line 77: define NH, SH

[Figure]

Line 78: shows that CCl4 is still being emitted...

Line 96: ...to 0.12-0.18 Gg yr-1, a decline of 5% yr-1

Line 106: Emission estimates by Hu et al. were...

Line 110: please state the Xiao et al. European CCl4 emissions in Gg (referred to later in the text)

Line 111: this study....did not derive regional variations that likely occur across Line 128: AGAGE and affiliated stations

Line 131: oceanic air masses and occasionally by air masses from over Ireland, UK and continental Europe

Line 154: 20-day back trajectories

Line 156: define ERA

Linre 226: ...macro areas (acronyms given in Table 1)

Line 227: define SRR

Line 242: geo-referenced

Line 259: and/or chlor-alkali industry

Line 283: add 'Australian CCl4 emissions are declining at 5% yr-1 (Fraser et al., 2014)

Line 545: a priori (blue squares)

Comments

Line 136: CCl4 is measured at MHD by GC-MS and GC-ECD - the latter data are preferred because there are inherent problems in AGAGE in measuring CCl4 by GCMS. Do these problems exist for GC-MS at JFJ, and, if they do, do they impact on this analysis

Line 165: a priori emissions. I suggest the following prior could be used - the Xiao et al. European emissions should be released according to the E-PRTR distribution of industrial emissions. Hu et al. (2016) showed conclusively the US emissions of CCl4 (and presumably European emissions of CCl4) are not significantly related to population distributions but are related to the distribution of chemical industrial activity. Why bias your prior in the likely wrong direction using largely (96%) population distributed emissions. This could lead to a significantly better a priori.

Line 260: this study and Hu et al. show that the CCl4 emissions are coming from industrial chemical hot-spots and are not related to population distributions. Landfills and domestic bleach sources tend to follow population distributions and these studies therefore tend to down-play land fills and domestic bleach as significant sources although tentative, I think this important conclusion can be made.

3.2.4 Comparison with NAME: why not run the NAME inversion using all 3 observation sites not just MHD?

Line 270 - Figure 6 compares UK and NWEU emissions of CCl4 with the latter significantly higher. At this point it would be instructive to compare the relative size of the chemical industries in these two regions - for example compare their chlor-alkali productions.

Line 284: per capita emissions. Since it has been shown that CCl4 emission distributions do not follow population distributions, then something better than per capita emissions could be calculated as a reference indicator, such as CCl4 emissions per unit of chemical production. I have done this for Hu et al USA emissions and Fraser et al. Australian emissions, as a function of chloro-alkali production - USA (0.39 kg CCl4/tonne Cl and Australia (0.41 kg CCl4/tonne/Cl). European Cl production numbers are available - it would be interesting to see what the European CCl4/Cl emission factor is.

---

## Referee Comment (RC2) · Anonymous Referee #2 · 10 Jun 2016

In this paper the authors take long-term atmospheric data records of carbon tetrachloride (CCl4) from several European sites and use an inversion-type model to derive top-down estimates of European industrial emissions. The rationale behind this is that there is a well-known discrepancy between global bottom-up emission estimates provided by producers and recent top-down estimates derived from atmospheric measurements. Assuming that the atmospheric loss processes are understood correctly, this discrepancy implies there is a missing or unknown source of this important ozone depleting chemical, which would explain why the global atmospheric abundance of CCl4 is not declining as fast as would be expected.

The major finding of this work is that Europe is a relatively minor contributor (4%) to global CCl4 emissions and that there are continuing small emission sources in particular regions of Europe. Although this is not a startling discovery (indeed a European

contribution of 4% was also calculated for the earlier period 1996-2004 by Xiao et al., 2010) this work is a useful contribution to the global jigsaw. Although I am not qualified to comment on the detailed mathematics that goes into the Bayesian statistics and inverse modelling, the uses of these methods by the authors and others have been widely published in recent years so I would expect them to be robust. The measurement and modelling teams both have an excellent reputation and I see no major flaws in their work. The manuscript is generally well written and I would recommend publication following consideration of the following minor comments.

Detailed comments

At the end of the discussion section (or in the conclusions) when putting their European emissions into a global context I wonder if the authors could summarise the current state of play regarding the CCl4 story. It seems from the references given that recent US, European, Australian and even Chinese top-down emissions still may still not add up to the total amount required to maintain the current atmospheric abundance. Is there still a missing source, or is the budget balanced within the various levels of uncertainty?

Line 35: "European emissions correspond to 4.0% of global emissions for 2006-2012". Do the authors mean cumulative emissions over the 7 year period or is it an average of 4% each year? Please clarify.

Line 61: "total chlorine in the troposphere". Firstly, do the authors mean total organic chlorine? Secondly, which part of the troposphere are they referring to?

Line 88: replace of with for i.e. "being responsible for more than half"

Line 136: I am intrigued as to why the MHD data is taken from the GC-ECD instrument rather than the Medusa-GC-MS that also operates at the site? At JFJ they are using data from a similar (identical?) Medusa GC-MS instrument, but the authors choose not to report the GC-MS data from MHD. Is there a reason for this?

Line 143: delete "are" i.e. "tertiary tanks used as"

Line 144: "at least twice" is not the same as "regularly calibrated". The working tanks are prepared at SIO and are calibrated (at SIO) at the beginning and end of the life of the tank?

Line 167: Is there a reference or web link for the E-PRTR database?

Line 185: move "also" – i.e. "During 2010-2014 data from JFJ were also used"

There are a few places in the text when the word "the" should be added:

Line 200: the emission distribution . . . Line 202: the main deviations . . . Line 210: during the study period . . . Line 240: The maps in . . . Line 259: the chloro-alkali industry . . .

Line 215-216: how can emissions be negative? Please explain in the text.

Line 216: is there a reference for the UNEP production database?

Line 217-218: "Such discrepancy holds also . . ." This sentence doesn't really make sense. Please explain and expand.

Line 225: change to ".. contribution to the total European emissions of CCl4 from the . . .."

Line 230: change to ". . .. as the main emitter in the EGD over the entire study period . . ."

Line 231: exactly 25%? Perhaps approximately 25% would be better (or give the range)

Line 233: replace "the" with "a" – i.e. "reached a maximum . . .."

Line 318: delete "as reported above"

Figure 2:

What actually are the time series? Are they averaged in any way or are they individual samples?

[Figure]

What is the dip seen in the CCl4 concentrations at CMN in 2006? This does not appear to be seen at MHD which suggests it is a local phenomenon. How can such a drop below the expected NH background be accounted for? The period seems to last for several months so presumably cannot be put down to a stratospheric event or southern hemisphere air. Does this period of abnormally low concentrations have any impact on the inversions? Why is the baseline signal in the middle panel so much more variable than the other 2? Perhaps this also relates back to the choice of ECD over MS at Mace Head?

Figure 4:

What do the error bars in Panel a represent? Please add an explanation in the caption. Figure 7:

I am not entirely convinced by the regression lines in Figure 7 or the trends described in lines 280-284. Could the yellow trend line be biased by the slightly higher value in 2006 and the slightly lower value in 2012? With the very large uncertainties highlighted by the error bars can you really say there is a statistically significant difference between the blue and yellow lines? It is hard to see from this Figure as they are plotted on different y axes. Can the authors say with any certainty that European emissions were falling faster than global emissions over this period?

---

## Referee Comment (RC3) · Anonymous Referee #3 · 19 Jun 2016

This well-written manuscript provides a valuable constraint regarding the discrepancy between global inventories and top-down estimates for CCl4 emissions. The authors present their atmospheric measurements, and a priori and posterior inversion results clearly and concisely, with adequate references to previous and related work.

With these European data added to a growing list of regional studies for CCl4 emissions, e.g., Hu et al. 2016, we are closer to understanding and attributing this emission discrepancy. The 'emission hot-spots' identified by this analysis beckons further work to collaborate these findings, and to seek additional evidence of the potential for missed sources in the landfill and chlor-alkali industries.

I recommend that this manuscript be published as is.

---

## Author Comment (AC1) · 2 Aug 2016

Response to reviewers

The authors would like to thank the reviewers for the very useful suggestions that allowed us to improve the robustness of our estimates.

Main changes:

The use of an alternative a priori did not produce significant differences in the emission estimates. Updated emission fluxes have been reported whenever applicable. However, we believe that the various tests performed contributed in improving the robustness of the results, as shown in the Supplementary Material detailed description and as also reported in the main text.

[Figure]

The detailed analysis of CCl4 emission factors required the inclusion of a new section (3.3).

Figures numbering has been changed according to the revised text. In addition, Figures 4 and 6 (current numbering) have been updated to comply with the revised text and results.

We made all the suggested changes along the text.

In the following we provide detailed answers to the specific comments.

Reviewer #1

CCl4 is measured at MHD by GC-MS and GC-ECD - the latter data are preferred because there are inherent problems in AGAGE in measuring CCl4 by GCMS. Do these problems exist for GC-MS at JFJ, and, if they do, do they impact on this analysis. Reply: We want to use as much receptors as possible in order to improve the model performance. However, in this case we found the following: we run the inversion removing JFJ time series and we found a difference in the a posteriori for the whole EGD < 5%. At the macro-area level this difference is relevant for CH macro-area only (>35%) but, being the contribution of CH very small (1.5% of the EGD emissions), this difference is smoothed when considering the whole EGD. This is probably due to the fact that at JFJ the CCl4 GC-MS time series is quite noisy and therefore, in this specific case, the signal do not contribute significantly to the inversion. Also the overlapping of the footprint of CMN and JFJ can be another cause. We also tested the GC-MS time series at MHD, finding a significant increase (40% higher) in emissions, and the correlation between the modelled and observed time series decreased from r2 0.72 to 0.2. The important role of MHD time series in the inversion results is due to the fact that the noisy GC-MS time series at MHD is not balanced by any nearby receptor. Finally, please note that at CMN we use a GC-MS system but this differs from the GC-MS system used at JFJ and MHD in the trapping temperature and GC column, giving a better signal to noise ratio.
Line 165: a priori emissions. I suggest the following prior could be used - the Xiao et al. European emissions should be released according to the E-PRTR distribution of industrial emissions. Hu et al. (2016) showed conclusively the US emissions of CCl4 (and presumably European emissions of CCl4) are not significantly related to population distributions but are related to the distribution of chemical industrial activity. Why bias your prior in the likely wrong direction using largely (96%) population distributed emissions. This could lead to a significantly better a priori. Reply: Thanks for this advice. Following your suggestion, we tested alternative a priori emission fields. The detailed description of the a priori emission field tested is provided in the Supplementary Material, along with an evaluation of their performance. After these tests we have chosen an a priori that, as the reviewer suggested, takes into account the distribution of industrial emission. This new approach produced a posteriori EGD emission values very similar to that obtained with the reference a priori previously used, but with better correlation values. Some differences are found at the macro-area level. In the revised manuscript all the emission values and trend have been updated, as well as the discussion (when necessary).

Line 260: this study and Hu et al. show that the CCl4 emissions are coming from industrial chemical hot-spots and are not related to population distributions. Landfills and domestic bleach sources tend to follow population distributions and these studies therefore tend to down-play landfills and domestic bleach as significant sources although tentative, I think this important conclusion can be made. Reply: The results obtained using the revised approach showed that this is the case in the EGD too and we added a statement on this. In addition, we would like to point out that the industrial activities in the E-PRTR also include emissions from landfills.

3.2.4 Comparison with NAME: why not run the NAME inversion using all 3 observation sites not just MHD? Reply: The meteorology available to the NAME model is not sufficiently high resolution over the Alps and northern Italian mountains to accurately capture the flow in these challenging areas. Meteorology of the order of a km would be

required to enable a reasonable estimate of the flow and therefore be of benefit to the inversion system.

Line 270 - Figure 6 compares UK and NWEU emissions of CCl4 with the latter significantly higher. At this point it would be instructive to compare the relative size of the chemical industries in these two regions - for example compare their chlor-alkali productions Reply: We didn't find any correlation between the size of chemical industries and the CCl4 emission fluxes, suggesting that it is not possible to identify an emission factor applicable to all industries. This is the reason why we didn't provide any graphical information.

Line 284: per capita emissions. Since it has been shown that CCl4 emission distributions do not follow population distributions, then something better than per capita emissions could be calculated as a reference indicator, such as CCl4 emissions per unit of chemical production. I have done this for Hu et al USA emissions and Fraser et al. Australian emissions, as a function of chloro-alkali production - USA (0.39 kg CCl4/tonne Cl and Australia (0.41 kg CCl4/tonne/Cl). European Cl production numbers are available - it would be interesting to see what the European CCl4/Cl emission factor is. Reply: thank you for your suggestion, we have modified the analysis and the text accordingly. In particular we moved the paragraph with the comparison with global emission trend to section 3.2.1. Section 3.3 now reports a detailed analysis of industrial emission factors.
* * *
Reviewer #2

At the end of the discussion section (or in the conclusions) when putting their European emissions into a global context I wonder if the authors could summarise the current state of play regarding the CCl4 story. It seems from the references given that recent US, European, Australian and even Chinese top-down emissions still may still not add up to the total amount required to maintain the current atmospheric abundance. Is there

still a missing source, or is the budget balanced within the various levels of uncertainty? Reply: Beside some still missing emission sources, there is also a re-consideration on the CCl4 lifetime. A new study by Butler at al. has been published on ACP while the present paper was under revision. This reconsideration narrowed the gap between top-down and bottom-up estimates. We added a reference to this paper and the related findings.

Line 35: "European emissions correspond to 4.0% of global emissions for 2006-2012". Do the authors mean cumulative emissions over the 7-year period or is it an average of 4% each year? Please clarify. Reply: It is the average on the period. We modified the text to make it clearer

Line 61: "total chlorine in the troposphere". Firstly, do the authors mean total organic chlorine? Secondly, which part of the troposphere are they referring to? Reply: Yes it is total organic chlorine in the whole troposphere. We added this information in the text.

Line 136: I am intrigued as to why the MHD data is taken from the GC-ECD instrument rather than the Medusa-GC-MS that also operates at the site? At JFJ they are using data from a similar (identical?) Medusa GC-MS instrument, but the authors choose not to report the GC-MS data from MHD. Is there a reason for this? Reply: when measuring CCl4, the performance of the GC-ECD is better than the Medusa system. Therefore, when available, GC-ECD data are used. For a detailed explanation of the implications for the inversion see reply to reviewer 1.

Line 144: "at least twice" is not the same as "regularly calibrated". The working tanks are prepared at SIO and are calibrated (at SIO) at the beginning and end of the life of the tank? Reply: yes this was a mistake. Tanks are calibrated at the beginning and at the end of life. We deleted "regularly".

Line 167: Is there a reference or web link for the E-PRTR database? Reply: yes, we added it in the text

Line 215-216: how can emissions be negative? Please explain in the text. Reply: we added the definition in the text

Line 216: is there a reference for the UNEP production database? Reply: Yes we added it in the text

Line 217-218: "Such discrepancy holds also . . ." This sentence doesn't really make sense. Please explain and expand. Reply: Done

Figure 2: What actually are the time series? Are they averaged in any way or are they individual samples Reply: The time series are raw data divided into baseline (black) and polluted (red)

What is the dip seen in the CCl4 concentrations at CMN in 2006? This does not appear to be seen at MHD which suggests it is a local phenomenon. How can such a drop below the expected NH background be accounted for? The period seems to last for several months so presumably cannot be put down to a stratospheric event or southern hemisphere air. Does this period of abnormally low concentrations have any impact on the inversions? Reply: we cannot explain the deep but data have not been flagged because we do not have any instrumental reason to flag them. However it should be noted that the inversion procedure is more affected by the extent of the enhancement above the baseline rather than the baseline absolute value.

Why is the baseline signal in the middle panel so much more variable than the other 2? Perhaps this also relates back to the choice of ECD over MS at Mace Head? Reply: Yes, see reply above

Figure 4: What do the error bars in Panel a represent? Please add an explanation in the caption. Reply: the error bar represents the uncertainty of the emission estimates. The uncertainty is found by the inversion routine, as described in the Supplementary material. The information is added in the caption

Figure 7: I am not entirely convinced by the regression lines in Figure 7 or the trends

described in lines 280-284. Could the yellow trend line be biased by the slightly higher value in 2006 and the slightly lower value in 2012? With the very large uncertainties highlighted by the error bars can you really say there is a statistically significant difference between the blue and yellow lines? It is hard to see from this Figure as they are plotted on different y axes. Can the authors say with any certainty that European emissions were falling faster than global emissions over this period? Reply: We reported two different trends: one for the entire period in which we run the inversion (2006-2014, red line) and one for the period for which we can make a comparison with the Global trend (2006-2012, blue line). We agree that the caption is not clear and we modified it. We agree with the reviewer that over the period 2006-2012 a statistically significant difference between the EGD emission estimates and the global ones cannot be detected. Moreover, the use of the alternative a priori emission field suggested by Reviewer# 1 produced a trend for the EGD emissions closer to the global one.

Please also note the supplement to this comment:
http://www.atmos-chem-phys-discuss.net/acp-2016-326/acp-2016-326-AC1-supplement.pdf
* * *
[Figure]

**Supplement:**

[revised manuscript text omitted]

Figure 4S shows the average percent contribution to the total CCl₄ flux for each macro area in the EGD over the period 2007-2014. FR alone results to be responsible for 65% of the emissions from the EGD, with an average emission of 0.02 Gg yr⁻¹. BE-NE-LUX and UK-IE follows with 13.5 and

750   11%, respectively. NEE, SCA and CH do not report any emission.

As reported in the paper main text (paragraph "Emission hot spots"), the inversion results estimate a CCl₄ emission flux much larger than that declared in the E-PRTR. For major detail, we report in Fig. 5S the percent ratio between emissions reported in the E-PRTR and our estimates for each

755   macro-area in the domain during 2007-2014. The E-PRTR reported emissions from the EGD represent on average, over the considered period, 4 % of the emissions obtained through the inversion. Lower discrepancies are found for the BE-NE-LUX and FR macro areas where the declared emissions reach the 30 % and 21% of inversion estimation, respectively.

760

[Figure]

**Figure 4S. Average percent contribution to the total CCl₄ flux for each macro area in the EGD over the period 2007-2014.**

[Figure]

**Figure 5S. Percent ratio between emissions as in the E-PRTR and the inversion results for each macro-area in the domain during 2007-2014.**

770 *Chlorine industry in Europe: Eurochlor*

One of the main $CCl_4$ emission source is the chlor-alkali industry. Information on chlorine and chlorine derivatives production in Europe is given by Eurochlor, an association representing the 97% of chlorine production in Europe. The total number of plants reported by Eurochlor over the period 2006-2014 is 84 (of which 10 in common with the E-PRTR). Eurochlor releases annual

775 reports where potential chlorine production for each industry is given, together with the adopted technology. However, Eurochlor reports do not include information on the $CCl_4$ emission factors according to the adopted technology. Since 1990's chlorine production in Europe is significantly changed. In 1997 ca 64% of the chlor-alkali industry was based on mercury cell technology and only 10% was based on the cleaner membrane cell process. Currently, the latest accounts by 60%

780 against the 25% of the mercury technology. Over the same period the use of diaphragm cells was reduced from 22 to 12%, while other technologies represent only the 2-3% of the total. Further uncertainties could be due to the employment of $CCl_4$ in industrial processes where it is used as a process agent in the chlor-alkali plants for the elimination of nitrogen trichloride and the recovery of chlorine from tail gases. In Europe plants that are allowed to use directly $CCl_4$ (European Union,

785 2010) were only eight in 2010, of which three in France. However, this source is difficult to assess because the allowed facilities do not have any obligation to report the actual use of the allocated $CCl_4$ quota and/or the transfer of this quota to another plant. According to DG CLIMA (2012) in 2011 only three chlor-alkali plants in Europe were using $CCl_4$, and reported emissions ranged from 0 to 30 g $CCl_4$/tonne annual chlorine capacity, depending on the frequency of use and the

790 occurrence of accidents (Brinkmann et al., 2014).

The graph in Figure 6.S reports the percent potential chlorine production for each macro area. The major contributor is the DE-AT macro area accounting for 40%, followed by BE-NE-LUX with 15.8%. FR, that according to the E-PRTR is responsible for the 65% of European $CCl_4$ emissions, is the third potential chlorine producer, accounting for the 12%.

795

[Figure]

**Figure 6S. Percent potential chlorine production for each macro area in the EGD (Eurochlor, 2014).**

800    In Figure 7.S the percentages of adopted technologies in each macro area in the EGD are reported. It should be noted that in ES-PT, CH and NEE more than 50% of the production is still based on the mercury cell technology. In FR and DE-AT there is still a significant use of diaphragm cells. To be noted that within the DE-AT macro area, Eurochlor does not report any plant in Austria.

[Figure]

805

**Figure 7S. Percentages of adopted technologies in each macro area. Hg: mercury cell technology; D: diaphragm cell; M: membrane cell (Eurochlor, 2014).**

810    *A priori* **emission field**

The construction of the *a priori* emission field is a challenging aspect of the methodology adopted in this study, since $CCl_4$ emission fluxes are affected by high uncertainty.

Possible CCl$_4$ emission sources are: chlor-alkali plants (UNEP, 2012); emissions produced by feedstock use; petrol-chemical, pesticide, and fire extinguisher industry (UNEP, 2006; 2012); methane chlorination, toxic waste treatment, landfills, incinerators (Fraser et al., 2014); and bleach containing domestic cleaning agents (Odabasi et al., 2008; 2014).

The latest have been evaluated up to 0.49 Gg yr$^{-1}$ for a population of 600 millions in the EGD and this amount has been distributed following the population (CIESIN, 2010) density in all the *a priori* emission fields tested in this work. The remaining non-diffuse emissions have been parameterised following six different ways (F1-F6).

A reference CCl$_4$ emission value for Europe is that given by Xiao et al. (2010), who estimated an average emission of 3.0 ± 1.6 Gg yr$^{-1}$ over 1996-2004. Since our tests have been performed for the year 2012, we applied to such average value a 2% decrease, following the projection given by Fraser et al. (2012), resulting in an emission from the EGD of 2.38 Gg yr$^{-1}$.

Here we give a detailed description of each *a priori* emission field tested: Please note that fluxes taken by Eurochlor and E-PRTR are always geo-referenced:

F1: 0.49 Gg yr$^{-1}$ distributed according to the population density; 1.89 Gg yr$^{-1}$ (i.e., the total estimated EGD emission of 2.38 Gg yr$^{-1}$ minus the 0.49 Gg yr$^{-1}$ diffuse emission) are attributed to the chloro-alkali plants evenly distributed among each single plant given in the E-PRTR and in the Eurochlor databases.

F2: 0.49 Gg yr$^{-1}$ distributed according to the population density; 1.89 Gg yr$^{-1}$ as follows: 50% of this flux is evenly distributed among each of the 37 plants listed in the E-PRTR and 50% evenly distributed among the Eurochlor plants (74); in this way F2 assigns a greater role to the E-PRTR plants.

F3: as for F2 but the 50% attributed to the E-PRTR plants is distributed according percent relative contribution to emissions declared by each plant (i.e., if a plant is declaring the 20% of the total CCl$_4$ reported in the E-PRTR, we assign to this plant the same percentage); similarly, for the Eurochlor plants the 50% is distributed according to the percent relative distribution of the declared chlorine production.

F4: the emissions declared in the E-PRTR, i.e. 0.064 Gg yr$^{-1}$ have been distributed among the single cells where plants are located (Eurochlor plants not included because no information about CCl$_4$ emissions is given by this database), while 2,32 Gg yr$^{-1}$ (i.e. the total estimated EGD emission of 2.38 Gg yr$^{-1}$ minus the 0.064 Gg yr$^{-1}$ E-PRT flux) are distributed according to the population density.

In addition, we have tested two *a priori* emission fields where the total EGD emission do not correspond to the 2.38 Gg yr$^{-1}$ derived from Xiao et al. (2010):

F5: 0.49 Gg yr$^{-1}$ distributed according to the population density; to such value a flux is added calculated applying an emission factor of 0.4 kg CCl$_4$ for each tonne of chlorine produced by all plants included in the Eurochlor database. Finally the CCl$_4$ emissions as declared in the E-PRTR are added. It should be noted that the *a priori* emission flux derived corresponds to 4.4 Gg yr$^{-1}$ for the EGD in 2012. The 0.4 kg CCl$_4$ for each tonne of chlorine produced emission factor has been suggested by Paul Fraser (personal communication).

F6: as in F5, but applying an emission factor of 0.03 kg CCl$_4$ for each tonne of chlorine produced by all plants (Brinkmann et al., 2014) included in the Eurochlor database. It should be noted that the *a priori* emission flux derived corresponds to 0.6 Gg yr$^{-1}$ for the EGD in 2012.

In order to evaluate the inversion performance for the various *a priori* emission fields tested, we compared i) the correlation values ($ra^2$) between the modelled and the observed concentration time obtained using the *a priori* emission fluxes F1÷F6; ii) the correlation values ($rb^2$) between the modelled and the observed concentration time obtained using the *a posteriori* emission fluxes F1÷F6. In Table 1S the $ra^2$ and $rb^2$ values for each station are reported as well as the emission flux produced by a given *a priori* emission field (F1÷F6) from the entire EGD. In all the reported tests, $rb^2$ values are always higher than $ra^2$ values, i.e. the *a posteriori* emission field gives account of a better representation of the variance of the measured signal with respect to the *a priori*. In addition, "EGD emission" *a posteriori* values obtained using quite different *a priori* emission fields are very similar (well within the error bar), confirming that the inversion is robust enough and converges towards a reliable emission estimate. In light of such results, we decided to use an "ensemble" *a priori* emission field, built as follows: to each macro area we assigned an emission flux given by the average, for that macro area, of the *a posteriori* emission fields produced by F1÷F6. The share given by diffuse emission has been distributed according to the population density, whereas the remaining share has been equally assigned (and geo-referenced) to each plant in that macro area. The "Ensemble" row in Table 1S reports the $ra^2$ and $rb^2$ values, as well as the obtained EGD emission flux. The "Ensemble" *a priori* emission field has been used for estimating $CCl_4$ emissions over the study period.

**Table 1S. Comparison among different *a priori* emission fields. F1÷F6 and "Ensemble" represent the *a priori* emission fields described in the text. $ra^2$: Correlation between modelled concentration fluxes obtained by a given *a priori* emission field and the observations at the three measurement sites. $rb^2$, as $ra^2$ but for the *a posteriori* emission fluxes. The tests have been performed for year 2012.**

| A priori emission field | CMN | | JFJ | | MHD | | EGD emissions |
|---|---|---|---|---|---|---|---|
| | $ra^2$ | $rb^2$ | $ra^2$ | $rb^2$ | $ra^2$ | $rb^2$ | Gg yr$^{-1}$ |
| F1 | 0.45 | 0.58 | 0.31 | 0.42 | 0.66 | 0.79 | 2.3 ± 0.8 |
| F2 | 0.45 | 0.58 | 0.29 | 0.39 | 0.66 | 0.79 | 2.1 ± 0.8 |
| F3 | 0.40 | 0.58 | 0.27 | 0.37 | 0.59 | 0.78 | 2.2 ± 0.8 |
| F4 | 0.48 | 0.58 | 0.35 | 0.44 | 0.70 | 0.78 | 2.1 ± 0.7 |
| F5 | 0.38 | 0.58 | 0.25 | 0.37 | 0.35 | 0.78 | 2.4 ± 0.9 |
| F6 | 0.49 | 0.58 | 0.32 | 0.42 | 0.74 | 0.79 | 2.1 ± 0.8 |
| | | | | | | | |
| Ensemble | 0.49 | 0.58 | 0.35 | 0.44 | 0.75 | 0.79 | 2.3 ± 0.8 |

**Subsets of data**

[revised manuscript text omitted]

CIESIN, Center for International Earth Science Information Network (CIESIN): Gridded Population of the World: Future Estimates, Socioeconomic Data and Applications Center (SEDAC), Columbia University, Palisades, NY, USA, 2010. Available at http://sedac.ciesin.columbia.edu/gpw.

EC DG CLIMA (European Commission, Directorate General Climate Action), Report of undertakings on consumption and emissions of controlled substances as process agents under regulation EC/1005/2009, 2012.

European Union, Commission Decision of 18 June 2010 on the use of controlled substances as process agents under Article 8(4) of Regulation (EC) No 1005/2009 of the European Parliament and of the Council, 2010/372/EU, 2010.

Fang X. K., Thompson R. L., Saito T., Yokouchi Y., Kim J., Li S., Kim K. R., Park S., Graziosi F., Stohl, A.: Sulfur hexafluoride (SF6) emissions in East Asia determined by inverse modelling, Atmos. Chem. Phys., 14, 4779-4791, 2014.

Fraser, P., Dunse, B., Manning, A. J., Wang, R., Krummel, P., Steele, P., Porter, L., Allison, C., O'Doherty, S., Simmonds, P., Mühle, J., and Prinn, R.: Australian carbon tetrachloride (CCl4) emissions in a global context, Environ. Chem., 11, 77–88, 2014.

975    Graziosi, F., Arduini, J., Furlani, F., Giostra, U., Kuijpers, L. J. M., Montzka, S. A., Miller, B. R., O'Doherty, S. J., Stohl, A., Bonasoni, P., and Maione, M.: European emissions of HCFC-22 based on eleven years of high frequency atmospheric measurements and a Bayesian inversion method, Atmos. Environ., 112, 196, 2015.

980    Keller, C. A., Brunner, D., Henne, S., Vollmer, M. K., O'Doherty, S., and Reimann, S.: Evidence for under-reported western European emissions of the potent greenhouse gas HFC-23, Geophys. Res. Lett., 38, L15808, doi:10.1029/2011gl047976, 2011.

Mahowald, N. M., Prinn, R. G., and Rasch, P. J.: Deducing $CCl_3F$ emissions using an inverse
985    method and chemical transport models with assimilated winds, J. Geophys. Res., 102, 28153–28168, 1997.

Maione M., Graziosi, F., Arduini, J., Furlani, F., Giostra, U., Blake, D.R., Bonasoni, P., Fang, X., Montzka, S.A., O'Doherty, S.J., Reimann, S., Stohl, A., and Vollmer, M.K.: Estimates of European
990    emissions of methyl chloroform using a Bayesian inversion method, Atmos. Chem. and Phys. 14, 9755-9770, doi:10.5194/acp-14-9755-2014, 2014.

Odabasi M.: Halogenated volatile organic compounds from the use of chlorine-bleach containing household products, Environ. Sci. and Technol. 42, 1445-1451, 2008.
995

Odabasi M., Elbir T., Dumanoglu Y., & Sofuoglu SC.: Halogenated volatile organic compounds in chlorine-bleach-containing household products and implications for their use, Atmos. Environ., 92, 376-383, 2014.

1000    Stohl, A., Hittenberger, M., and Wotawa, G.: Validation of the Lagrangian particle dispersion model FLEXPART against large scale tracer experiment data, Atmos. Environ., 32, 4245–4264, 1998.

Stohl, A., Forster, C., Frank, A., Seibert, P., and Wotawa, G.: The Lagrangian particle dispersion
1005    model FLEXPART version 6.2, Atmos. Chem. Phys., 5(9), 2461–2474, 2005.

Stohl, A., Seibert, P., Arduini,J., Eckhardt, S., Fraser, P., Greally, B. R., Lunder, C., Maione, M., Mühle, J., O'Doherty ,S., Prinn, R. G., Reimann, S., Saito, T., Schmidbauer, N., Simmonds, P. G., Vollmer, M. K., Weiss, R. F., and Yokouchi, Y.: An analytical inversion method for determining
1010    regional and global emissions of greenhouse gases: Sensitivity studies and application to halocarbons, Atmos. Chem. and Phys. 9, 1597-1620, 2009.

UNEP/CTOC, United Nations Environment Programme/Chemicals Technical Options Committee, 2006. Report of the Chemicals Technical Options Committee (Nairobi, Kenya).
1015

UNEP/TEAP, United Nations Environment Programme/Technology and Economic Assessment Panel, 2012. Report on the Technology and Economic Assessment Panel (Nairobi, Kenya).

Xiao, X., Prinn, R. G., Fraser, P. J., Weiss, R. F., Simmonds, P. G., O'Doherty, S., Miller, B. R.,
1020    Salameh, P. K., Harth, C. M., Krummel, P. B., Golombek, A., Porter, L. W., Butler, J. H., Elkins, J. W., Dutton, G. S., Hall, B. D., Steele, L. P., Wang, R. H. J., and Cunnold, D. M.: Atmospheric three-dimensional inverse modelling of regional industrial emissions and global oceanic uptake of carbon tetrachloride, Atmos. Chem. Phys., 10, 10421-10434, doi: 10.5194/acp-10-10421-2010, 2010.
1025

---

## Author Response (AR2)

Urbino, September 6th, 2016

Dear Prof Martyn Chipperfield,

Following your suggestions we included in the text (Section 3.1 Time Series Statistical Analysis) further information, previously given in response to referees only, concerning the use of the GC-MS-Medusa data at JFJ and the dip in the CMN time series (both shown in Figure 2).

Hoping that this revision will satisfy your request and thanking you for your help, we send our best wishes.

Michela Maione